

# Investigating the level of agreement of two positioning protocols when using dual energy X-ray absorptiometry in the assessment of body composition

Flinn Shiel[1], Carl Persson[1], Vini Simas[2], James Furness[1,2], Mike Climstein[2,3] and Ben Schram[1,2]

[1] Department of Physiotherapy, Faculty of Health Sciences & Medicine, Bond University, Robina, Queensland, Australia
[2] Water Based Research Unit, Faculty of Health Sciences & Medicine, Bond University, Robina, Queensland, Australia
[3] Exercise Health & Performance Faculty Research Group, The University of Sydney, Lidcombe, New South Wales, Australia

Corresponding author
James Furness, jfurness@bond.edu.au

## ABSTRACT

**Background.** Dual energy X-ray absorptiometry (DXA) is a commonly used instrument for analysing segmental body composition (BC). The information from the scan guides the clinician in the treatment of conditions such as obesity and can be used to monitor recovery of lean mass following injury. Two commonly used DXA positioning protocols have been identified—the Nana positioning protocol and the National Health and Nutrition Examination Survey (NHANES). Both protocols have been shown to be reliable. However, only one study has assessed the level of agreement between the protocols and ascertained the participants' preference of protocol based upon comfort. Given the paucity of research in the field and the growing use of DXA in both healthy and pathological populations further research determining the most appropriate positioning protocol is warranted. Therefore, the aims of this study were to assess the level of agreement between results from the NHANES protocol and Nana protocol, and the participants' preference of protocol based on comfort.

**Methods.** Thirty healthy participants (15 males, 15 females, aged 23–59 years) volunteered to participate in this study. These participants underwent two whole body DXA scans in a single morning (Nana positioning protocol and NHANES positioning protocol), in a randomised order. Each participant attended for scanning wearing minimal clothing and having fasted overnight, refrained from exercise in the past 24 h and voided their bladders. Level of agreement, comparing NAHNES to Nana protocol was assessed using an intra-class correlation coefficient (ICC), concordance correlation coefficient (CCC) and percentage change in mean. Limit of agreement comparing the two protocols were assessed using plots, mean difference and confidence limits. Participants were asked to indicate the protocol they found most comfortable.

**Results.** When assessing level of agreement between protocols both the ICC and CCC scores were very high and ranged from 0.987 to 0.997 for whole body composition, indicating excellent agreement between the Nana and NHANES protocols. Regional analysis (arms, legs, trunk) ICC scores, ranged between 0.966 and 0.996, CCC ranged between 0.964 and 0.997, change in mean percentage ranged between −0.58% and 0.37% which indicated a very high level of agreement. Limit of agreement analysis

using mean difference ranged between −0.223 and 0.686 kg and 95% CL produced results ranging between −1.262 kg and 1.630 kg. The majority (80%) of participants found the NHANES positioning protocol more comfortable.

**Discussion**. This study reveals a strong level of agreement as illustrated by high ICC's and CCC's between the positioning protocols, however systematic bias within limit of agreement plot and a large difference in 95% confidence limits indicates that the protocols should not be interchanged when assessing an individual. The NHANES protocol affords greater participant comfort.

# INTRODUCTION

Tissue composition assessment and analysis is commonly undertaken by using dual-energy X-ray absorptiometry (DXA) (*Nana et al., 2012*). The need for a device to accurately and reliably measure bone mineral density as an indicator of an individual's bone health, led to the development and implementation of the DXA scanner (*Lewiecki, 2005*). Dual energy X-ray absorptiometry emits energy sources that are absorbed at different degrees of attenuation relative to the type of tissue they encounter; thus enabling clear imagining of different tissues (fat mass, lean mass and bone) based upon the distinctive elements of these tissues (*Rothney et al., 2009*). Due to these distinct properties of measurement, the DXA scan calculates an individual's total body composition (BC), together with an individual's regional BC; thus, the DXA is a popular instrument in research and clinical settings. Furthermore, DXA produces 0.004 mSv of radiation in each BC scan, equating to less than 1% of the maximum radiation dosage of 5 mSv in a year, as described by Australian Radiation Protection and Nuclear Safety Agency (*ARPANSA, 2005*). Therefore, the minimal level of radiation from DXA scans enables researchers and clinicians to widely use this instrument to assess BC on a regular basis. Research drawn from BC scans has assisted clinicians and researchers to further their understanding of a number of conditions, including obesity and undernourished individuals (*Lee & Gallagher, 2008*). When applying BC scanning to athletes, it has been identified that those with higher muscle mass in pre-season, have a decreased likelihood of suffering bone related injuries during the season (*Georgeson et al., 2011*). Nevertheless, it is important to note that the DXA's reliability must be ascertained prior to statistical data being extracted, analysed and applied within a clinical and or sporting population.

In previous studies a variety of statistical analysis methods have been undertaken including intra-class correlation coefficients (ICC), percentage change and Pearson correlations to assess the reliability of the DXA, all of which have found DXA to be reliable (*Bilsborough et al., 2014*; *Climstein et al., 2015*; *Colyer et al., 2016*; *Covey, Berry & Hacker, 2010*; *Covey et al., 2008*; *Kerr et al., 2016*; *Lohman et al., 2009*; *Moon et al., 2013*; *Nana et al., 2012*; *Nana et al., 2013*; *Smith-Ryan et al., 2017*). However higher reliability

is found in studies that account for biological and technical errors, especially the use of a reproducible positioning protocol. The National Centre for Health Statistics, National Health and Nutrition Examination Survey (NHANES) body composition positioning protocol (*NHANES, 2013*) and the Nana positioning protocol, founded by Alisa Nana, are the two most popularly used protocols (*Nana et al., 2012*). It is important to note the Australian and New Zealand Bone Mineral Society (ANZBMS) employs the same body position as the NHANES positioning protocol.

F Shiel et al. (2017, unpublished data) have systematically assessed studies using the Nana and NHANES positioning protocols and concluded that there is a high level of evidence and excellent reliability for the Nana positioning protocol, and a moderate level of evidence but excellent reliability for the NHANES, and therefore the Nana protocol should be considered the gold standard for BC DXA scanning. *Kerr et al. (2016)* is the only study to date which has compared the Nana and NHANES positioning protocols, concluding that the Nana protocol's reliability is superior in assessment of regional BC, fat mass (FM) and bone mineral content (BMC). This study also recommended that positioning protocols should not be interchanged, and proposes that the Nana positioning protocol is more comfortable for the participant (*Kerr et al., 2016*). However, it should be noted that the Kerr study has used modified versions of the original protocols, which may have altered the participants perceived comfort level during the scan.

As such the primary aim of our study is to conduct an independent comparison of the Nana and NHANES positioning protocols in terms of results and level of agreement. The finding of this research will either strengthen the findings suggesting the Nana protocol produces superior results or increase the level of evidence for the NHANES protocol. Additionally, this study aimed to assess which of the two main positioning protocols identified in the published literature is more comfortable.

## METHODS

### Study overview

During a single session, each participant underwent a total body scan twice, being repositioned between each scan. The two scans consisted of one using the Nana positioning protocol, with feet and hands positioned in radio-opaque pads; the other scan utilized the NHANES positioning protocol scan, where the hands are positioned faced down on the scanning bed. The order of the positioning protocol scans was randomised. Each participant was asked to choose which positioning protocol, Nana or NHANES, was the most comfortable, and why they selected that positioning protocol.

### Participants

Fifteen males and fifteen females ($n = 30$) were recruited from the local university and the greater public to partake in this comparative study. Thirty participants were selected based upon the previously published recommendations for reliability studies (*Lexell & Downham, 2005*). Participants underwent an anthropometrical analysis of height (to the closest 0.1 cm) using a medical stadiometer (Harpenden, Holtain Limited, Crymych, UK) and mass (to the closest 0.1 kg) on medical scales (WM202, Wedderburn, Bilinga,

**Table 1  Participant characteristics.**

|  | Males ($n = 15$) | Females ($n = 15$) | Group ($n = 30$) |
|---|---|---|---|
| Age (yr) | $27.8 \pm 7.2$ | $31.3 \pm 11.9$ | $29.6 \pm 10.1$ |
| Height (cm) | $178.7 \pm 7.3$ | $164.7 \pm 8.9$ | $171.7 \pm 10.7$ |
| Mass (kg) | $78.9 \pm 8.8$ | $62.4 \pm 9.7$ | $70.6 \pm 12.4$ |

Australia) prior to undergoing a BC scan on the DXA. Participant characteristics can be found in Table 1. Prior to partaking in the study, all participants were informed of the testing procedures and signed a consent form. The study was granted ethics approval by Bond University Human Research Ethics Committee (RO15221).

## Standardised baseline conditions

On the morning of the scan, the participant confirmed they had fasted overnight; rested and refrained from strenuous exercise for the previous 24 h; wore minimal clothing (males: underwear, females: underwear, sports bra or two piece bathers); bladder voided; as well as jewellery and metal removed, prior to scanning.

## DXA instrument

BC was measured using a narrow angle fan beam Lunar Prodigy DXA machine (GE Healthcare, Madison, WI, USA) with automatic analysis performed using GE enCore 2016 software (GE Healthcare). DXA provides three-component approximation of bone tissue and soft tissue (lean tissue i.e., muscle) and fat tissue (ANZBMS, 2014). The DXA was calibrated daily prior to any scans using a phantom as per manufacturer's guidelines. The machine used for the study has previously been found to produce very high reliability for BMD (0.998), lean mass (0.989) and fat mass (0.995) (Climstein et al., 2015).

## Standardised DXA operational protocol

All scans were performed by the same licensed researcher with all scans analysed automatically by the GE enCORE 2016 software. Two BC protocols were utilised, the NHANES positioning protocol and the Nana positioning protocol (Fig. 1). The NHANES protocol required the participant to be positioned in a supine position in the middle of the densitometry table with head straight, space between the arms and torso, palms flat on the table, and feet together secured by a strap (NHANES, 2013). When utilising the Nana positioning protocol, participants were centrally aligned in the scanning area with their feet placed in a custom-made foam block to maintain a consistent distance between the subject's feet (15 cm) in each scan. The custom-made foot blocks were made from styrofoam and were transparent under the DXA scan. Additionally, the subject's hands were placed in custom-made foam and plastic paddles to ensure a mid-prone position with a standardised gap (3 cm) between the palms and trunk. These hand paddles created minimal changes to the scan analysis (Nana et al., 2012). Additionally, a strap around the ankles was utilised as per the NHANES protocol, to ensure that the only difference between protocols was the positioning block/paddles.

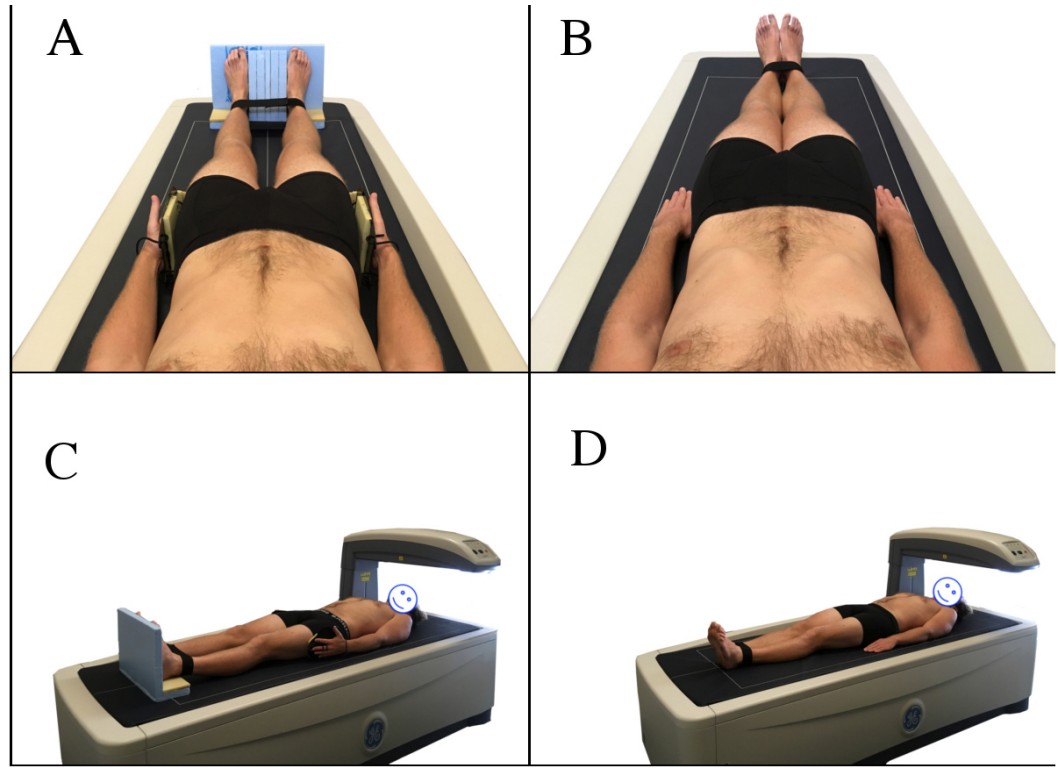

**Figure 1** Nana positioning protocol (A, C) and NHANES positioning protocol (B, D).

## Statistical analysis

IBM SPSS (version 24.0) and a custom reliability spreadsheet from Sportscience web site (http://www.sportsci.org) were used to analyse the data. Anthropometrical data were presented as means and standardised deviations. IBM SPSS 24 was utilised to assess Intra-class Correlation Coefficient (3, 1) with Confidence Intervals (CI), Concordance Correlation Coefficient (CCC) with 95% Confidence Limits (CL) and create Limit of Agreement analysis plots and assess mean difference and associated confidence limits. This specific ICC was selected based on the published work of *Trevethan (2016)*. Percentage change in mean and typical error expressed as coefficient of variation as a percentage (CV%) were calculated using the customised Sportscience spreadsheet.

## RESULTS

All results comparing the Nana positioning protocol with the NHANES positioning protocol (Fig. 2) are presented in Table 2. When assessing the BC using two different positioning protocols; the results of the whole body (tissue, FM, LM and BMC) scans and all regional (arms, legs and trunk) scans were excellent based on ICC's and percentage change in mean statistics. The results are also illustrated in the Limit of Agreement analysis plots for whole body (Fig. 3) and Table 3 for all regions.
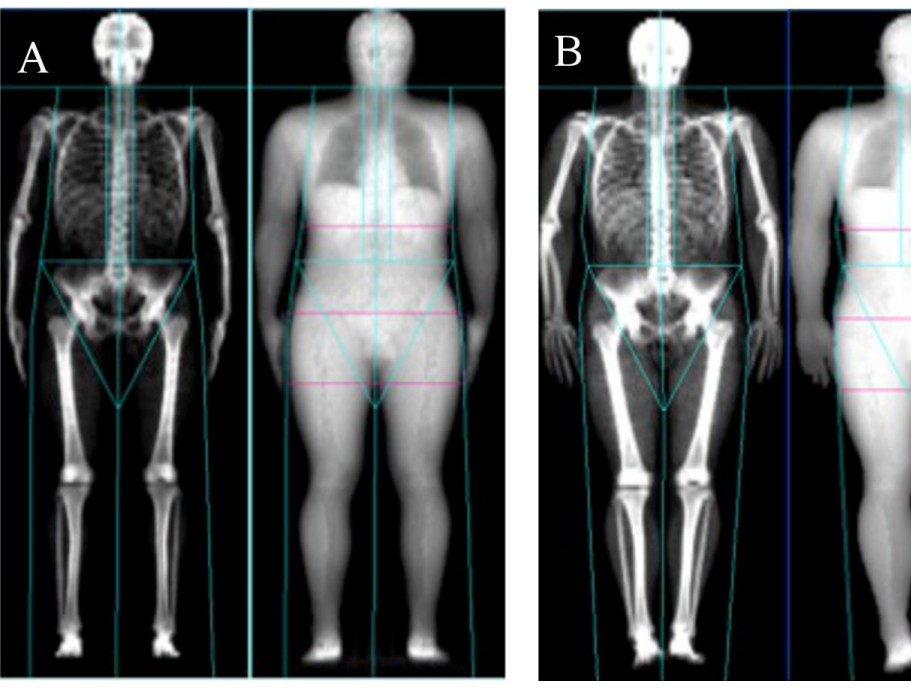

**Figure 2**  Nana positioning protocol (A) and NHANES positioning protocol (B).

**Table 2**  Level of agreement between Nana vs NHANES positioning protocols.

| | | % Δ in mean | Typical error as CV% | ICC | CI (95%) | CCC | CL (95%) |
|---|---|---|---|---|---|---|---|
| Whole body | Tissue | −0.47 | 0.10 | 0.987 | 0.970–0.994 | 0.987 | 0.976–0.993 |
| | Fat | 0.21 | 0.30 | 0.997 | 0.992–0.999 | 0.997 | 0.994–0.998 |
| | Lean | −0.68 | 0.32 | 0.997 | 0.905 - 0.999 | 0.997 | 0.995–0.998 |
| | BMC | 0.06 | 0.03 | 0.990 | 0.586–0.998 | 0.989 | 0.983–0.994 |
| Arms | Tissue | −0.32 | 0.19 | 0.982 | 0.745–0.995 | 0.982 | 0.968–0.989 |
| | Fat | 0.08 | 0.13 | 0.966 | 0.923–0.984 | 0.964 | 0.936–0.980 |
| | Lean | −0.39 | 0.15 | 0.980 | 0.329–0.996 | 0.980 | 0.966–0.980 |
| | BMC | 0.01 | 0.01 | 0.979 | 0.876–0.993 | 0.994 | 0.989–0.997 |
| Legs | Tissue | −0.58 | 0.38 | 0.984 | 0.822–0.995 | 0.983 | 0.971–0.990 |
| | Fat | −0.10 | 0.19 | 0.992 | 0.983–0.996 | 0.992 | 0.986–0.996 |
| | Lean | −0.49 | 0.30 | 0.987 | 0.837–0.996 | 0.987 | 0.977–0.992 |
| | BMC | 0.02 | 0.01 | 0.996 | 0.795–0.999 | 0.997 | 0.998–0.999 |
| Trunk | Tissue | 0.37 | 0.42 | 0.993 | 0.977–0.997 | 0.993 | 0.987–0.996 |
| | Fat | 0.22 | 0.29 | 0.991 | 0.975–0.996 | 0.991 | 0.983–0.995 |
| | Lean | 0.18 | 0.39 | 0.993 | 0.986–0.997 | 0.993 | 0.988–0.996 |
| | BMC | 0.02 | 0.02 | 0.973 | 0.841–0.991 | 0.972 | 0.951–0.984 |

**Notes.**

% Δ in Mean, percentage change in mean; CV, confidence variance; ICC, intra-class correlation coefficient; CI, confidence interval; CCC, concordance correlation coefficient; CL, confidence limit.

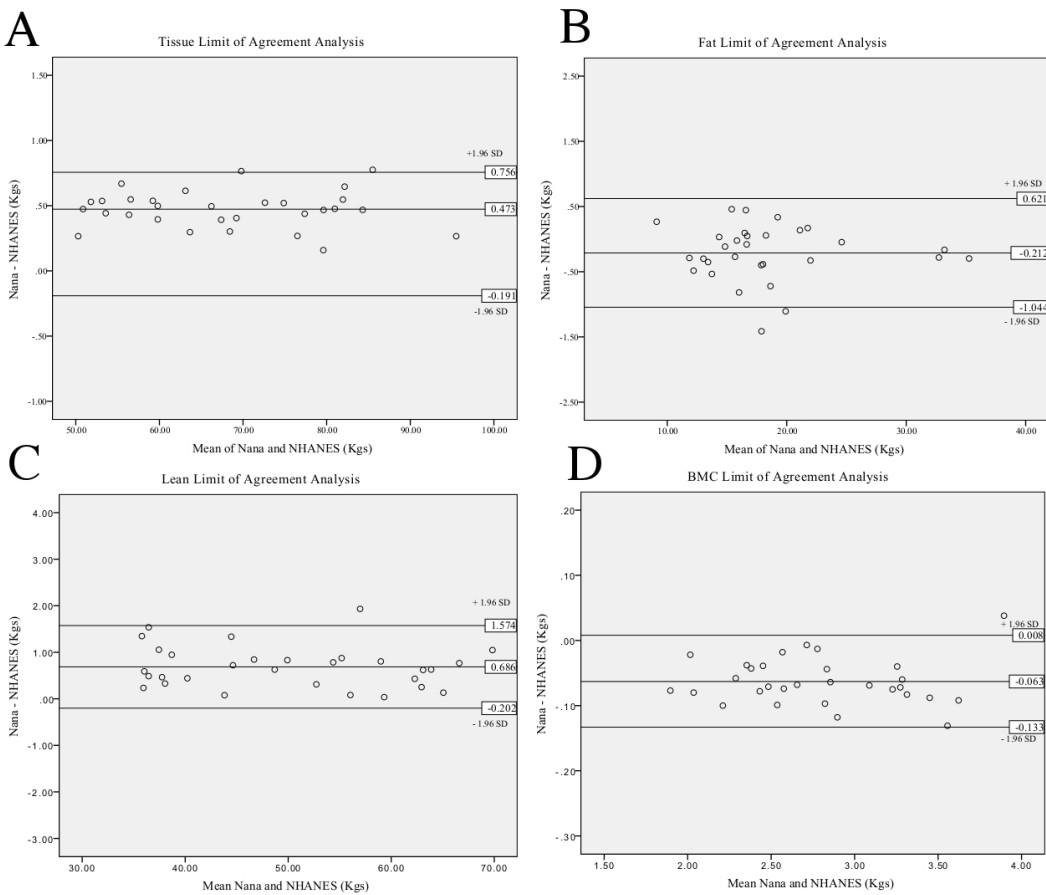

**Figure 3** **Limit of Agreement analysis for Nana versus NHANES whole body positioning protocols.**
Tissue analysis (A), fat analysis (B), lean analysis (C), BMC analysis (D).

Percentage change in mean when comparing the two protocols has produced results that range between −0.68% and 0.37%. Trunk was the regional area with the smallest variance of the four sites (whole body, arms, legs and trunk) as described in Table 2, with results ranging from 0.02% to 0.37%. Whole body scans produced the largest variance, with results ranging from −0.68% to 0.21%.

The typical error expressed as CV% of the agreement between the positioning protocols produced results ranging between 0.01% and 0.42%. The parameter of BMC was assessed to produce the smallest typical error across the four different sites (whole body, arms, legs and trunk). The tissue parameter was found to be the highest in three of four assessment sites (arms, legs and trunk).

A very high level of agreement between the two positioning protocols is evident through an ICC ranging between 0.966 and 0.999. Whole body fat mass-produced the highest ICC of 0.997, with a 95% CI [0.992–0.999]. The fat mass of the arms produced the lowest ICC of 0.966, with a 95% CI [0.923–0.984].

**Table 3  Limit of agreement between Nana vs NHANES positioning protocols.**

|  |  | Mean difference | Lower CL | Upper CL |
|---|---|---|---|---|
| Whole body | Tissue | 0.473 | −0.191 | 0.756 |
|  | Fat | −0.212 | −0.621 | 1.044 |
|  | Lean | 0.686 | 0.202 | 1.574 |
|  | BMC | −0.063 | −0.133 | 0.008 |
| Arms | Tissue | 0.321 | 0.193 | 0.836 |
|  | Fat | −0.074 | −0.432 | 0.283 |
|  | Lean | 0.396 | 0.014 | 0.807 |
|  | BMC | 0.000 | −0.020 | 0.021 |
| Legs | Tissue | 0.586 | 0.458 | 1.630 |
|  | Fat | 0.099 | 0.420 | 0.618 |
|  | Lean | 0.488 | 0.350 | 1.327 |
|  | BMC | −0.005 | −0.030 | 0.020 |
| Trunk | Tissue | 0.366 | 0.806 | 1.538 |
|  | Fat | −0.223 | −1.017 | 0.572 |
|  | Lean | −0.176 | −1.262 | 0.911 |
|  | BMC | −0.022 | −0.071 | 0.027 |

**Notes.**
CL, Confidence Limit (95%).

Additional to the ICC, the CCC illustrates very good results with the results ranging between 0.964 and 0.997. The whole body lean mass produced the highest result of 0.997 with 95% CL of 0.995–0.998. Similar to the ICC result the fat mass of the arms produced the lowest correlation of 0.964 with 95% CL 0.936–0.980.

Limit of Agreement analysis plots (Fig. 3) for the whole body reveal a bias between the two measures when assessing tissue as the zero value lies outside of the interval. This indicates that the Nana protocol consistently produced larger values than the NHANES protocol. Limit of agreement analysis using mean difference between the protocols ranged between −0.223 and 0.686 kg across the parameters with arm measures the smallest difference. The 95% CL produced results ranging from −1.262 kg for the lower limit up to 1.630 kg for the upper limit. All mean differences fell within the define CL except for the leg fat assessment.

When questioned about which protocol was the more comfortable, 24 out of 30 participants (80.0%) chose the NHANES positioning protocol as the more comfortable of the two protocols assessed.

## DISCUSSION

The primary aim of this study was to focus upon technical error associated with positioning and establish the level of agreement between the two identified positioning protocols. This study also sought to identify which DXA scan positioning protocol was the more comfortable for participants. In this study, we conducted all scans of BC using a Lunar DXA machine, located at Bond Institute of Health & Sport. To minimise the chance of technical error, one licensed researcher (qualified through ANZBMS) conducted

all thirty scans as recommended for reliability studies (*Lexell & Downham, 2005*). To further decrease the chance of error affecting the results, biological factors such as time of day of scanning, hydration, exercise and food metabolism have been identified and accounted for.

This study found that the level of agreement between the Nana and the NHANES positioning protocols was very high when using a variety of statistics including percentage change in mean, accompanied with typical error, or ICC, accompanied with CI. The percentage change in mean findings of this study for the whole body (tissue −0.47%, FM 0.21%, LM −0.68%, BMC 0.06%) is similar to the results of the previous study comparing the two protocols (tissue −0.4%, FM −2.8%, LM 0.3%, BMC −0.7%) (*Kerr et al., 2016*). The results of this study suggest that the level of agreement between the two protocols when doing regional analysis is also very good however these results are opposed to previously published research that conclude there is a large difference between protocol results (*Kerr et al., 2016*).

The assessed percentage change in mean in this study is smaller across the all parameters assessed except for whole body tissue mass in comparison to the only other study that has compared the two positioning protocols (*Kerr et al., 2016*). This may be due to the stringent methodology used in our study. As both studies have accounted for biological factors creating errors the source of difference can only be technical error. As such in this study, the NHANES protocol was followed as prescribed in NHANES Body Composition Procedures Manual 2013 (*NHANES, 2013*). The participant's feet were secured together with a strap and the hands were placed in a pronated position (palms down on the table), reducing the likelihood of movement artifacts. In comparison, the previous research conducted by Kerr and colleagues, the legs were secured with a strap but positioned a significant distance apart, possibly allowing for small amounts of internal rotation and adduction as these movements were not limited. Furthermore, the hands were held in a neutral position, possibly allowing for small rotational movements. The combination of these two adjustments to the prescribed NHANES positioning protocol could possibly have created movement artifacts and altered results.

This is the first study to use an ICC to assess the level of agreement between the two positioning protocols. Very high ICC results are deemed to be between 0.90 and 1.00 (*Munro & Visintainer, 2005*), and our results (0.996–0.999) fall within this described range. Additionally, the concordance correlation results (0.964–0.997) coupled with the ICC results indicated that the level of agreement between the two positioning protocols is very high, however this needs to be coupled with the mean difference and confidence limits analysis before deciding if the protocols are interchangeable.

The limits of agreement between the two positioning protocols when plotted into limit of agreement analysis plots (Fig. 3) reveals a systematic bias in the parameter of whole body tissue. The systematic bias illustrates that the Nana protocol consistently produces higher results than the NHANES protocol, possibly due to the use of the foam blocks used to secure the feet. Additionally, Table 3 reveals that the mean difference lies outside of the defined 95% confidence limits for the leg fat parameter, this is due to this parameter having a large difference between the standard deviation and the mean when comparing

the protocols. Applying the limit of agreement findings clinically illustrates a large variance, for example if the participant's lean mass was 50 kg and mean difference 1.75 kg then this equates to 4% change. These factors indicate that the two positioning protocols should not be used interchangeably even though the ICC results are very high.

When assessing which positioning protocol (Nana or NHANES) was deemed the most comfortable; this study found that 24 out of 30 participants (80.0%) chose the NHANES positioning protocol to be the most comfortable; this result is in direct opposition to previous findings (*Kerr et al., 2016*). Upon closer inspection of the methods employed, it appears Kerr and colleagues altered the original NHANES and Nana positioning protocols, which would have affected the perceived comfort levels of participants. The modified version of the NHANES positioning protocol they employed, would have required muscular activation and control; therefore, decreasing the participant's perceived comfort. When using the Nana positioning protocol, a strap was added to the original Nana protocol, which secured the participants arms for approximately seven minutes during scanning; hence decreasing the muscular activation and increasing the participant's perceived comfort. In our study, the majority of participants who chose the NHANES as the most comfortable did so because they felt their hands and arms were in a more relaxed position.

The Nana positioning protocol, where the feet are placed in radio-opaque blocks to maintain plantargrade ankle position, allows for taller individuals to be scanned with a decreased risk of plantar flexion and the participant's feet moving outside the scanning field (*Nana et al., 2012*). Most individuals in our study over the height of 185 cm chose the Nana positioning protocol for comfort, and did so based on not having to actively maintain their foot in plantargrade during the scan. Additionally, the Nana positioning protocols' use of pads to maintain the hands in a midprone position allows for larger individuals (width wise) to be scanned more easily in comparison to the NHANES, where the individual's hands are pronated flat on the table.

Future research needs to investigate if certain positioning protocols are more applicable for different participants dependent upon their size. Furthermore, more research is required to ascertain the difference between the positioning protocols when using regional analysis.

The implications for clinical practice are that the decision of which positioning protocol to employ should be based on comfort, i.e., the size of the participant's and not purely on the level of evidence for the protocols as both protocols produce very good results. As such, the NHANES protocol should be the first choice when scanning based on the comfort findings, however the Nana protocol provides a fantastic alternative for larger individuals.

## CONCLUSION

When all sources of biological and technical errors have been accounted for, the Nana and NHANES positioning protocols both produce a very high level of agreement as demonstrated by very high results. However, the systematic bias revealed in the limit of agreement plot and the large 95% CL indicated that the two protocols should not be used interchangeably. Anecdotally, the NHANES positioning protocol was more comfortable.

## ACKNOWLEDGEMENTS

The authors would like to thank the 2017 graduating Physiotherapy cohort for donating their time for this study.

### Funding

The authors received no funding for this work.

### Competing Interests

The authors declare there are no competing interests.

### Author Contributions

- Flinn Shiel conceived and designed the experiments, performed the experiments, analyzed the data, contributed reagents/materials/analysis tools, wrote the paper, prepared figures and/or tables, reviewed drafts of the paper.
- Carl Persson conceived and designed the experiments, performed the experiments, analyzed the data, contributed reagents/materials/analysis tools, wrote the paper, reviewed drafts of the paper.
- Vini Simas performed the experiments, reviewed drafts of the paper.
- James Furness conceived and designed the experiments, contributed reagents/materials/analysis tools, reviewed drafts of the paper.
- Mike Climstein contributed reagents/materials/analysis tools, reviewed drafts of the paper.
- Ben Schram conceived and designed the experiments, reviewed drafts of the paper.

### Human Ethics

The following information was supplied relating to ethical approvals (i.e., approving body and any reference numbers):

The Bond University Human Research Ethics Committee granted approval for this study to take place. Approval number: RO15221.

### Data Availability

The raw data has been supplied as Data S1.

### Supplemental Information

Supplemental information for this article can be found online at http://dx.doi.org/10.7717/peerj.3880#supplemental-information.

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
