# Peer review of "Investigating the level of agreement of two positioning protocols when using dual energy X-ray absorptiometry in the assessment of body composition"

_PeerJ, doi:10.7717/peerj.3880_

## Round 0.1 · original submission · Major Revisions

The reviewer's have identified several issues that need to be addressed including, but not limited to, the current utilization of the "Nana protocol" amongst researchers and/or clinical practitioners. Please provide point by point response regarding each issue raised by each reviewer and, identify where specifically in the revised manuscript the issue was addressed.

Reviewer 1 ·

Basic reporting

The article was written well for the most part. However, there were some instances where phrasing and spelling could be improved (see below under General comments). Existing literature was discussed effectively, with limited previous investigations performed in this area. Raw data were mostly made available, however the findings related to participant comfort were not presented effectively and the clarity of graphs contained in Figures could be improved.

Experimental design

While the question seems novel it is not well developed in the Introduction section leading from discussion on previous research. Also, there is some confusion over the actual research question at times given “reliability” is referred to consistently and is not actually measured in the submitted article. The methodological approach regarding the scanning procedures and protocols was presented at a high level.

Validity of the findings

The findings are mostly presented in sufficient detail; however, some data analyses were not explained in the Methods and surprisingly appeared in the Results section. Further, the conclusion is unclear given a definitive recommendation based on the provided data was not provided and some findings were not discussed regarding differences between approaches. Some adequate explanations for various findings was presented in the Discussion.

Additional comments

Introduction
Line 86: “on regular” should be “on a regular”.
Lines 95-99: This is a one-sentence paragraph that doesn’t really develop an idea fully. I would suggest combining it with the following paragraph or elaborating on the studies presented to present some actual reliability data for supportive evidence.
Line 111: “moderate level” of what specifically? Are you referring to “evidence” here? Please specify.
Line 113: “is the only study” should be “completed the only study” or similar. Also, why is Shiel et al. not given with a year – is this the unpublished work? If so, probably no need to reference.
Lines 120-123: There needs to be a stronger link to this paragraph. You finish the previous paragraph by highlighting that research demonstrates Nana’s protocol is more reliable and comfortable, but then don’t set up the importance of your investigation. Further development to show the need for your study is needed here in 1-2 sentences I feel.

Methods
Line 131: “the” should be inserted before “scanning bed”.
Line 132: Was block randomisation followed whereby the same number of participants completed each scan first?
Line 137: “and” should be between males and females without a comma.
Line 139: This is confusing as your study doesn’t seem to be a reliability study but rather a comparative study between scanning protocols. Please make sure your terminology and direction match the actual research question throughout.
Line 152: “jewelry” is incorrectly spelled.
Line 155: Has this device been shown to be reliable and valid?
Line 182: “data was” should be “data were” given it is the plural form.

Results
Line 207: There is no mention of calculating the CV previously in the manuscript?
Table 2: Can “Whole body” please be presented horizontally rather than vertically?
Figure 3: The numbers of the axes are very hard to see in this figure – can it be reworked to be larger?
Overall: There was no detail regarding the participant rating for comfort – this is a key outcome that should be presented in your results somewhere in this section.

Discussion
Line 238: “This studies primary aim” should be “The primary aim of this study”.
Line 254: Please delete “studies regional analysis” and replace with “study” here.
Line 264: Please delete the second “as” in this line.
Line 268: How does the feet distance apart affect movement artefact? Some elaboration on this suggestion would strengthen your argument here.
Line 271-274: This paragraph doesn’t offer anything that is not already known in the manuscript. Further, reliability of the approaches were not assessed but rather the level of agreement between the two was. This needs to be made clear throughout. I would recommend elaborating on these results further to strengthen this paragraph and transmit an important outcome with explanation to the reader.
Line 301: “minuet” is misspelled.
Lines 300-304: This is not really a measured outcome of the study though and is a little off topic. I would consider removing this paragraph.
Line 308: What is meant by “the real difference”? Were your findings not “real” or are you referring to a more valid approach?
Line 311: But size was not factored into your approach so I don’t see how it should shape your clinical outcomes.
Line 312: So what approach is recommended? From your descriptions here it appears that you are recommending either can be used, however your Bland-Altman data showed the tissue data where consistently higher in the Nana than the NHANES. This is not discussed anywhere in this section. Also, in the Introduction you mention that the Nana has been shown to possess greater reliability, and given reliability was not measured in your study, shouldn’t existing reliability data be factored into your conclusions?

References
There are inconsistencies throughout the reference list that require addressing – e.g. some journals do not have capitals assigned to words, some article titles are given capitals for each word, & is used in some journal titles, etc.

·

Basic reporting

The manuscript is generally well written and clearly describes the study (although see comments below).

Fig 3 is unclear, the Y scale should be compressed and the axis titles and legends are uninformative (what does Mmeanlean actually mean?). The 2 SD limits should be stated. What are the units? Please redraw.

Tabulate LOA data for all body regions.

How widely is the Nana protocol used? The paper has been cited 52 times on Web of Science but it appears that citing papers are concerned with the main aim of the Nana pape, the effects of daily activities on DXA" rather than the use of the protocol. If it is not widely used then this point should be made. This should not be seen, however, to question the worth of the present paper.

Experimental design

The authors have generally described the methods adequately. However a few points require attention.

Was the survey of participant's protocol preference undertaken in a systematic manner? was a specific survey form used or simply anecdotal comment? Provide more details.

The authors used data automatically provided by the equipment software. This is OK but does need clarification, for example, what is exactly defined as "tissue". This is a generic term that is presumably being used in specific way by Lunar. Please explain.

The statistical analysis is generally acceptable but I question the correctness of the use of ICC for method comparison. I believe that Lin's concordance correlation is more appropriate. I suspect that you will find that Lin's r and ICC are very similar but statistical rigour should be applied. The authors also refer to Bland Altman plots. This is colloquial, it would be more appropriate to refer to limits of agreement analysis. Paired t tests can also be used to assess the significance of position differences (or combined for all body regions in an ANOVA to see if there is segment-position interaction).

Single scans for each position only were performed. This begs the question as to whether the inter-position differences are similar to intra-position differences and hence may not be "true" inter-position differences. This needs to be addressed.

Validity of the findings

The authors present limits of agreement analysis but fail to discuss adequately these findings. For example, are the positioning interchangeable? The mean difference in lean is small (Fig 3) but the LOA are about =/- 2 (kg?) for a mean lean mass of around 50 kg. This equates to a 4% difference. Is this clinically acceptable? Can you justify the final sentence of the Discussion in the Abstract? This implies exchangeability of methods.

The LOA plots are for whole body only. I suspect that the LOA may be wider for body segment data. This should be provided. Rather than additional plots a table of bias and LOA for all body regions could be supplied.

Discussion requires a strengths and weakness paragraph.

Additional comments

Minor points to note

Line 77 It would be more correct to refer to different degrees of attenuation rather than "rates".
Line 109. It would be better to phrase as " Shiel et al (unpublished data) have systematically...."
Line 181. The website has a number of spreadsheets - which one was used. Why was this used rather than SPSS?
Line 262,. It is unclear how the studies have accounted for biological error? Please explain.
Line 182 and elsewhere. "Data" is a plural, i.e. data were...

---

## Round 0.2 · Minor Revisions

Please address the additional minor issues that reviewers have noted. Please provide a point by point response that clarifies how and where in the manuscript the issues were addressed.

Reviewer 1 ·

Basic reporting

Elements of basic reporting in the submitted manuscript meet the journal standards.

Experimental design

Elements of experimental in the submitted manuscript meet the journal standards.

Validity of the findings

Elements of validity in the submitted manuscript meet the journal standards.

Additional comments

The authors should be commended on addressing the author comments thoroughly, and as a result producing an improved manuscript. I do however have some minor points that require consideration:

INTRODUCTION
Lines 107-109: This seems like a bit of a stretch regarding development of a need for your study. I think speculating that bias existed in previous work should not be included and that an alternative reason/explanation should be presented here to show more work is needed in this area. Perhaps some of the positioning issues raised in this study might be useful.

DISCUSSION/CONCLUSION
Can you make a recommendation using the collective evidence from past research and your findings, considering aspects such as reliability, validity, comfort, etc.? You conclude that the protocols should not be used interchangeably, but which is ultimately recommended?

·

Basic reporting

The authors have satisfactorily amended the manuscript in response to queries raised in my original review.

Experimental design

The authors have satisfactorily amended the manuscript in response to queries raised in my original review. I accept the author's arguments for the use of ICC although I still believe that concordance correlation rather than ICC is preferable for the type of analysis described here.

Validity of the findings

The authors have satisfactorily amended the manuscript in response to queries raised in my original review.

Additional comments

The authors have satisfactorily amended the manuscript in response to queries raised in my original review.

---

## Round 0.3 · accepted · Accept

The manuscript has been revised sufficiently to be accepted for publication.

Reviewer 1 ·

Basic reporting

No comment

Experimental design

No comment

Validity of the findings

No comment

Additional comments

The authors have sufficiently addressed all suggested revisions. Thanks for the opportunity to review this revised version.